# Unsupervised Domain Adaption for High-Resolution Coastal Land Cover Mapping with Category-Space Constrained Adversarial Network

**Jifa Chen** [1], **Guojun Zhai** [1,2], **Gang Chen** [1,3,*], **Bo Fang** [1,3], **Ping Zhou** [1] **and Nan Yu** [4]

1. College of Marine Science and Technology, China University of Geosciences, Wuhan 430074, China; chenjifa@cug.edu.cn (J.C.); zhaiguojun@cug.edu.cn (G.Z.); fangbo@cug.edu.cn (B.F.); pingzhou@cug.edu.cn (P.Z.)
2. Naval Institute of Hydrographic Surveying and Charting, Tianjin 300061, China
3. Key Laboratory of Geological Survey and Evaluation of Ministry of Education, China University of Geosciences, Wuhan 430074, China
4. Hubei Key Laboratory of Marine Geological Resources, China University of Geosciences, Wuhan 430074, China; yunan@cug.edu.cn
* Correspondence: ddwhcg@cug.edu.cn; Tel.: +86-138-0713-4417

**Abstract:** Coastal land cover mapping (CLCM) across image domains presents a fundamental and challenging segmentation task. Although adversaries-based domain adaptation methods have been proposed to address this issue, they always implement distribution alignment via a global discriminator while ignoring the data structure. Additionally, the low inter-class variances and intricate spatial details of coastal objects may entail poor presentation. Therefore, this paper proposes a category-space constrained adversarial method to execute category-level adaptive CLCM. Focusing on the underlying category information, we introduce a category-level adversarial framework to align semantic features. We summarize two diverse strategies to extract category-wise domain labels for source and target domains, where the latter is driven by self-supervised learning. Meanwhile, we generalize the lightweight adaptation module to multiple levels across a robust baseline, aiming to fine-tune the features at different spatial scales. Furthermore, the self-supervised learning approach is also leveraged as an improvement strategy to optimize the result within segmented training. We examine our method on two converse adaptation tasks and compare them with other state-of-the-art models. The overall visualization results and evaluation metrics demonstrate that the proposed method achieves excellent performance in the domain adaptation CLCM with high-resolution remotely sensed images.

**Keywords:** coastal; land cover mapping; domain adaptation; category-wise; adversarial learning; self-supervised learning

## 1. Introduction

Coastal land cover mapping (CLCM) provides a detailed and intuitive presentation of ground objects in the land–sea interaction zone, which is the necessary and sufficient premise for land investigation, resource development, and eco-environment protection [1–3]. In the past decade, the continuous evolution of space and sensor technologies has made remote sensing enter into the Big Data era [4]. An intuitive advance is the favorable circumstance to achieve mass production of land cover while meeting large-scale and high-resolution needs. However, high-resolution remotely sensed (HRRS) images acquired in various scenarios are easily affected by irresistible factors, e.g., seasonal climates, regional conditions, and sensor models. Unfortunately, these discrepant factors may result in remarkable data divergences in the appearance distribution for scenes and ground objects. Therefore, it is yet a challenging task to achieve large-area and high-precision CLCM production automatically.

Recently, owing to the powerful capability of characterizing nonlinear features, deep convolutional neural networks have demonstrated a wide impact in image classification, pixel-level segmentation, and object recognition [5]. Numerous representative semantic segmentation models such as FCN [6], U-Net [7], PSPNet [8], and Deeplab systems [9–11] have been applied to conduct land cover mapping and have achieved superior performances. Despite this, such fully supervised approaches are extremely hungry for dense annotated data. Excellent models trained in one scene (source domain) will entail significant performance degradation when generalizing them to other scenarios (target domains). On the other hand, pixel-level manual labeling is prohibitively tedious and expensive. In such cases, developing a comprehensive algorithm that integrates domain shift to the land cover mapping becomes increasingly important.

As a special case of transfer learning, unsupervised domain adaption (UDA) has been taken to narrow the performance divergences introduced by the mismatch between the source and target domains [12]. Its core matter utilizes the unlabeled data from the target domain to circumvent the expensive annotation work as the labeled data are available in the source space. Along this line, the related UDA research mainly involves two branches. Conventional methods [13,14] leverage the manual-extracted features to minimize the domain gap. Another exploited workaround is carried out with deep learning technology, where the maximum mean discrepancy (MMD) [15,16] and adversarial framework [17,18] present the more common strategies. For the latter, adversarial learning seeks to minimize an approximate domain discrepancy distance through an adversarial objective to the domain discriminator. Besides, generative adversarial approaches [19] closely connected with them show a powerful image or feature generation capability, in which adversarial loss guides feature transformation from the source to the target domain.

Inspired by the advancement, it has become a popular trend to integrate domain adaptation and generative adversarial learning to conduct dense segmentation, and it has made fascinating achievements. FCNs in the wild [20] first employed adversarial learning to perform global alignment at feature-level, ultimately executing the adaptive segmentation work. After that, minimizing the distance of potential features between the data domains generalizes to a popular workaround [21–24]. Another research line focuses on pixel-level adaptation [25–27], aiming to address the domain shift problem by performing data augmentation in the target domain. These methods consistently expand from other extra image-to-image translation or style transfer frameworks [28,29]. For instance, CyCADA [25] transferred the source images to the target domain with pixel-level constraint via CycleGAN [28], followed by a feature alignment. Nevertheless, aligning the marginal distribution does not necessarily lead to satisfactory performance as there is no explicit constraint on the prediction in the target domain. Category-level adaptation [30–32] and entropy minimization [33,34] have recently been introduced to enhance the consistency of local semantics in the procedure of global alignment. Among them, CLAN [30] paid attention to the class-level joint distribution that made each class consistent with the adaptive adversarial loss. ADVENT [33] imposed entropy minimization to match data distribution by searching for weighted self-information in an adversarial flow. However, in general, adversarial adaptation always adjusts the global statistics with a naive discriminator, ignoring the underlying category information of the target domain. Even though FADA [35] proposed a fine-grained discriminator to alleviate this problem, its domain label from a single strategy may bring about adverse effects. Moreover, the majority of reports generally focus on segmentation work for diverse driving scenarios. Their limitation may give rise to performance degradation in high-resolution coastal scenes since the ground objects show remarkable low inter-class variances and multi-scale characteristics with intricate spatial details.

In this work, we propose a novel category-space constrained adversarial network (CsCANet) for cross-domain CLCM. The central formulation is to implement the feature alignment with generative adversarial learning while generalizing it to multiple levels via a robust baseline model. Specifically, the method consists of a segmentation module

and multiple adaptation modules, where the former is divided into feature extractor and pixel-level classifier. Following alternate training, the hybrid framework learns from the source space in a supervised way and transforms it into the target domain for our specific segmentation task. The primary contributions of this paper are summarized as follows:

1. Referring to the characteristics of HRRS images in coastal areas, we propose a category-level UDA approach to achieve land cover mapping across image domains, which emphasizes the advance of adversarial learning in generating and aligning the feature spaces.

2. For the category-level adaptation framework, we focus on the underlying category space of the target domain and introduce a category-wise discriminator to fine-tune the segmentation network. In light of heterogeneous situations, two different strategies are adopted to extract domain labels for the discriminator.

3. With the lower-level features concerning local details and higher-level ones encoding global context presentations, we integrate the adaptation modules with a similar architecture to each feature stack, aiming to align the semantic features at multiple spatial scales.

4. Experiments in two coastal datasets demonstrate that the proposed method enables the cross-domain CLCM to be realized and achieves excellent performance compared with other state-of-the-art models.

The remainder of this paper is arranged as follows. The background is reported in Section 2. Section 3 introduces our proposed method and presents its implementation details. Then, Section 4 describes the experimental procedures and results on two benchmark datasets, while we discuss the effectiveness of various designed modules in Section 5. Finally, our work and future research are concluded in Section 6.

## 2. Background

### 2.1. Adversarial Learning

Adversarial learning has recently become popular and has been explored in generative tasks since Goodfellow et al. [19] proposed the Generative Adversarial Nets (GAN) as a pioneering report. Adversarial learning essentially presents a dynamic mini-max game where the generative adversarial method is always divided into two antagonistic modes: a generative module *G* and a discriminative module *D*. Within the iterative training, *G* strives to generate imitative samples to deceive the discriminator by capturing the data distribution. Meanwhile, the target of *D* is to distinguish the generated distribution from real ones via a binary domain label. The whole process seeks *G* to minimize the divergence while updating *D* to maximize the separation, which can be formulated as follows:

$$G^* = \underset{G}{\arg\min}\underset{D}{\max}\left\{E_{x \sim P_{data}}[\log D(x)] + E_{x \sim P_G}[\log(1 - D(x))]\right\} \tag{1}$$

where $P_{data}$ and $P_G$, respectively, indicate the real and generated distribution.

The advance of GAN over other generative approaches is that there is no complex sampling and inference [17]. After that, numerous variants have served a breadth of visual tasks, e.g., image generation, style transfer, and image labeling. Using the deep fully convolution framework, DCGAN [36] provides pioneering guidance for complex mapping when CGAN [37] denotes an extension that makes it possible to link additional information such as the category relation of training samples. Presented as an excellent work, CycleGAN [28] performs the unpaired image–image translation by adopting the bi-directional consistency losses and adversarial losses. In summary, with its outstanding performance, the generative adversarial approach has now become a fundamental strategy for unsupervised domain adaptation.

### 2.2. Self-Supervised Learning

Even though deep supervised learning has made outstanding successes in the past decade, there is a fatal flaw with excessively relying on manual annotations. As an alternative,

self-supervised learning (SSL) adopts input data itself to dig supervision information for training. Precisely, SSL captures pseudo labels via a semi-automatic process or a partial prediction by leveraging the rest of the data. There are three summarized objective-based types: generative, contrastive, and generative-contrastive [38]. Generally, this strategy of SSL benefits various downstream tasks without the need for expensive supervision information.

Denoted as a branch of semi-supervised learning, SSL has been used in various image-related tasks [39–41]. For instance, a broad span of domain adaptation applications [35,41] leverage SSL to learn the decision boundary between source and target data. These approaches enable the promotion of the global feature matching of the different data domains while performing well in class-wise alignment. The SSL approach is also employed to carry out pixel-level annotating when the ground truth is not accessible. Under this scenario, the related methods [12,31] are often guided by the cross-entropy loss between the dense prediction and generated pseudo label. In our work, we leverage SSL to execute the pixel-level segmentation task in an unsupervised way. Note that the SSL strategy is simultaneously applied in our domain adaptation framework and segmentation module. First, we denote the dense predictions from the target data as the discriminator labels to update the adversarial adaptation network in training iterations. Second, the pseudo labels from the above adaptative predictions are regarded as the ground truth to fine-tune the segmentation network for target images. To a certain extent, SSL overcame the defect of missing annotations and has achieved distinguished contributions.

## 3. Materials and Methods

### 3.1. Problem Setting

Having access to the source image set with dense annotations and the target image set without any references (Figure 1a), we focus on the problem of unsupervised domain adaptation for the CLCM with HRRS images. The goal is to learn a pixel-level segmentation network in a supervised way and then achieve correct predictions for the target images in an unsupervised manner. Due to the divergence of marginal and joint distribution in both datasets (domains), deep convolutional models trained on source data always fail to generalize to the target space.

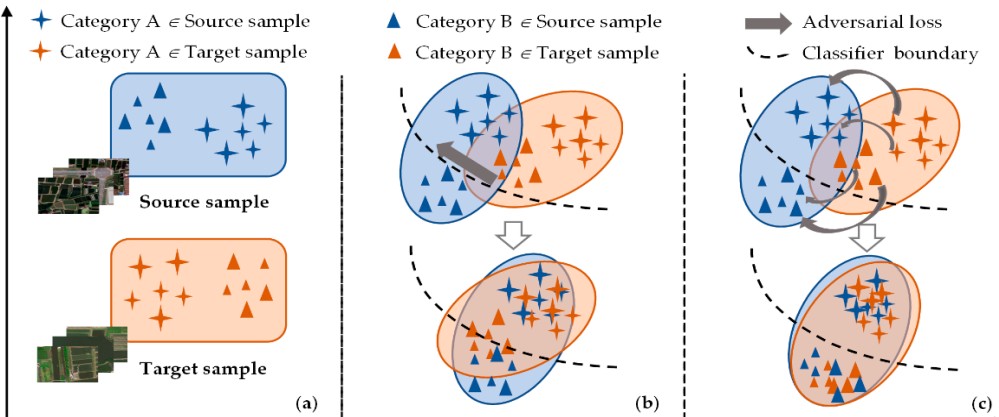

**Figure 1.** Illustration of conventional and our proposed unsupervised domain adaptation method. (**a**) Data samples from the source and target domain. (**b**) Conventional adversarial learning with global domain discriminator. (**c**) Our category-space constrained adversarial learning with multi-level category-wise discriminator.

To address the adverse effect of domain shift, we resort to a generative adversarial framework that learns the feature mapping between the source and target domain. Conventional adversaries-based methods commonly leverage the global discriminator as the domain judge, which only aligns the global marginal distribution. In such cases, it may lead to the misclassification of categories within the target domain (Figure 1b). Considering the underlying semantic structure, we fuse the category information into the multi-level

adversarial procedure by replacing the single naive discriminator with our multi-level category-wise discriminators. As illustrated in Figure 1c, this strategy implements the local domain matching for category features at multiple scales while performing the global domain alignment.

*3.2. Network Architecture*

3.2.1. Overall Formulation

The proposed CsCANet serves the cross-domain CLCM task via a multi-level adversarial framework and an extra self-supervised learning module (Figure 2). Specifically, the whole architecture comprises three fully convolutional networks: feature extractor $F$, pixel-level classifier $C$, and category-wise discriminators $D_i$, where $i = 1, 2, \ldots, n$, presents the level of adversarial adaptation scheme. Source images $X_S = \{X_S^i\}_{i=0}^{N^S}$ with annotated labels $Y_S = \{Y_S^i\}_{i=0}^{N^S}$ and target images $X_T = \{X_T^i\}_{i=0}^{N^T}$ are given as the inputs of the network, where $N^S$ and $N^T$ indicate the number of respective samples. Then, the shared baseline network $F$ generates multiple feature stacks (i.e., $F_i(X_S, \theta)$ and $F_i(X_T, \theta)$) for both the source and target domain at different spatial scales. Our target is to make the multi-level semantic features $F_i(X_S, \theta)$ and $F_i(X_T, \theta)$ close to each other. Hence, four category-wise discriminators with the analogous architecture are designed to achieve category-level domain adaptation for aligning the features in specific scales. Note that source label $Y_S$ and target prediction $C(F_i(X_T, \theta), \mu)$ are, respectively, denoted as the domain labels to discriminators. Furthermore, the prediction of the shared classifier $C$ is presented as the result of the CLCM, which also forwards to optimize the feature extractor and classifier via the segmentation losses.

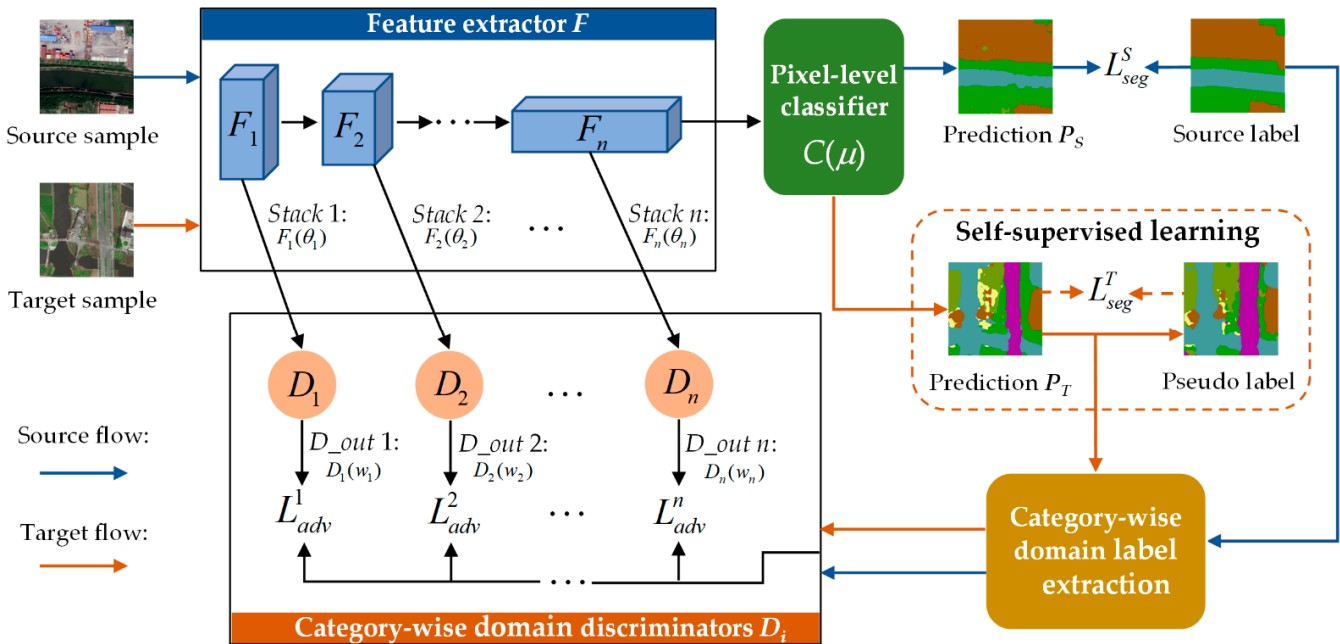

**Figure 2.** An overview of the proposed category-space constrained adversarial network for cross-domain coastal land cover mapping. The procedure is mainly composed of source flow and target flow that are drawn in different colors.

With the proposed network, the joint loss objective for the hybrid adaptation task can be formulated from two primary modules:

$$L = L_{seg}(X_S, X_T) + \lambda_{adv} L_{adv}(X_S, X_T) \qquad (2)$$

where $L_{seg}$ consisted of $L_{seg}^S$ and $L_{seg}^T$ denotes the cross-entropy losses between the prediction and label (truth or pseudo) in the source and target domains. $L_{adv}$ indicates the adversarial

losses that align with the category-wise data distribution. Besides, $\lambda_{adv}$ presents the weight coefficient to promote backward propagation steadily. For the target domain, the loss $L_{seg}^T$ is individually used for self-supervised learning to fine-tune the segmentation network toward better adaptation.

### 3.2.2. Domain Labels Extraction Module

For cross-domain dense segmentation, each image contains numerous pixels that represent multiple instances. Exploring adversarial learning for domain adaptation in the right way is a vital premise. The majority of global adaptation approaches adopt the single binary values (either "0" or "1") as the opposite domain labels (Figure 3a), ignoring the category-space constraints in the target domain. Our developed architecture introduces a category-wise discriminator for adaptively aligning the semantic features. For this purpose, extracting the category-wise domain labels denotes a crucial component module.

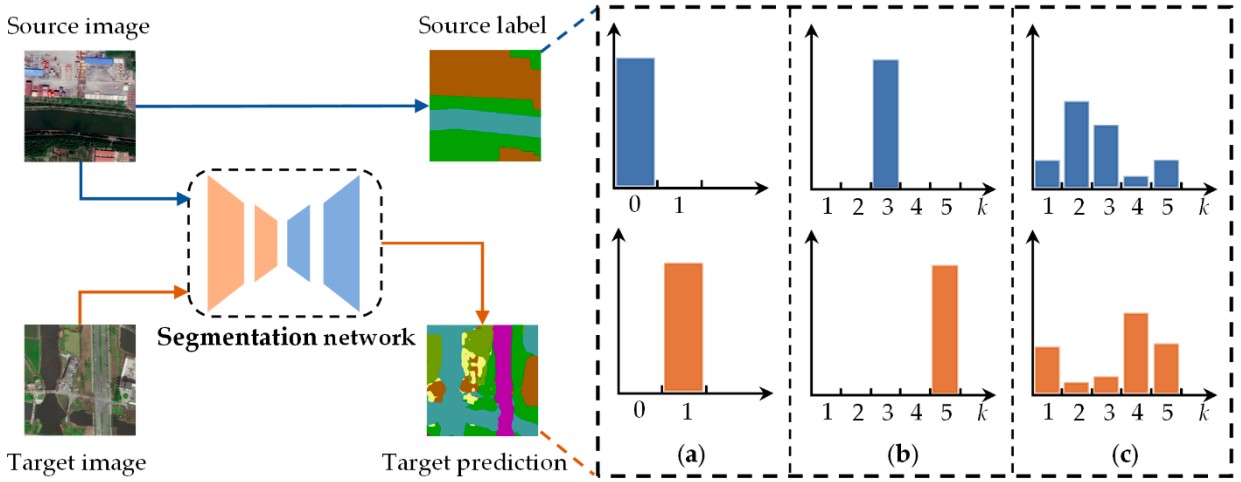

**Figure 3.** Illustration of domain label extraction module using different strategies. (**a**) Binary label. (**b**) Category-wise hard label. (**c**) Category-wise soft label. The elements drawn in blue present the source flow, while the orange ones indicate the target flow.

We seek the category information contained in both domains to construct the domain label for each sample. The constraint that the ground truth in the target domain is not accessible is contradictory to expect category-level alignment via the target category information. Referring to self-supervised learning, treating the target label as a learnable hidden variable is a feasible choice. We use the prediction of $C$ as the domain label to supervise the discriminator since the target domain shares the same semantic categories as the source data. In general, there are two summarized strategies for extracting domain labels, whose outcomes are divided into category-wise hard and soft labels. For the former shown in Figure 3b, executing the one-hot encoding is a straightforward solution, which can be formulated as follows:

$$hl^{(i,k)} = \begin{cases} 1, & k = \underset{k}{\mathrm{argmax}} P^{(i,k)} \\ 0, & others \end{cases} \tag{3}$$

where $i \in (N = H \times W)$ presents the pixel position, and $P^{(i,k)}$ gives the softmax probability prediction of the $k$th category. In this plan, we try to generate domain labels from the most confident predictions and hope that they are mostly correct. As a result, the method only adopts the category with the highest confidence for domain adaptation. This strategy enormously depends on the predicted outcomes of $C$.

Focusing on the adaptation problem to each category, we leverage the category-wise soft label as an alternative strategy (Figure 3c). Unlike the hard one, the latter utilizes

the probability prediction of all the channels to implement category-level adaptation, denoted as:

$$sl^{(j,k)} = \frac{\exp(P^{(j,k)})}{\sum_{k=1}^{K} \exp(P^{(j,k)})} \tag{4}$$

where $j \in (N = H \times W)$ presents the pixel position, and $P^{(j,k)}$ indicates the logits probability prediction of the $k$th category.

In our proposed architecture, we simultaneously employ two diverse strategies to extract category-wise domain labels instead of a single one. For the source data, the hard process is selected where we use the available ground truth as the domain label. In fact, this operation can offer the highest confidence rather than the probability prediction of $C$. On the other hand, we adopt the soft label from dense prediction per iteration to execute adversarial learning for the target domain. In essence, this belongs to a process of self-supervised learning.

### 3.2.3. Single-Level Adaptation Adversarial Framework

With the generative adversarial learning, the domain adaptation flow is generally executed by alternately updating the segmentation network $G$ and discriminator $D$. To be specific, $G$ is composed of feature extractor $F$ and pixel-level classifier $C$, where $G = (F \to C)$. Our single-level adaptation framework that focuses on the output last feature stack from $F$ also follows the above two procedures.

In our category-space constrained adaptation, we divide the output from $D$ and extracted domain labels into $k$ channels, aiming to encourage category-wise adversarial learning. It enables $D$ to model more complex underlying structures between categories. During the training iterations, $D$ is optimized to distinguish the features from cross domains. The training objective can be written as:

$$
\begin{aligned}
L_d(X_S, X_T) = \quad & -\sum_{i=1}^{N=H \times W} \sum_{k=1}^{K} (hl)^{(i,k)} \log(D(F(X_S)^{(i,k,0)}) \\
& -\sum_{j=1}^{N=H \times W} \sum_{k=1}^{K} (sl)^{(j,k)} \log(D(F(X_T)^{(j,k,1)})
\end{aligned}
\tag{5}
$$

where $hl^{(i,k)}$ and $sl^{(j,k)}$, respectively, indicate the label for source pixel $i$ and target pixel $j$.

As an antagonistic procedure, $G$ is trained with the segmentation loss $L_{seg}^S$ from the source domain and the adversarial loss $L_{adv}$ on the target space. This stage seeks to update $F$ and $C$ with the fixed $D$. We begin by defining the cross-entropy loss $L_{seg}^S$ to enforce the prediction close to the annotated ground truth, observed in:

$$L_{seg}^S(X_S) = -\sum_{i=1}^{N=H \times W} \sum_{k=1}^{K} (Y_S)^{(i,k)} \log((P_S)^{(i,k)}) \tag{6}$$

where $Y_S$ and $P_S$ denote the ground truth and dense predictions for source samples.

Second, under the assumption that we do not diverge far from the target solution, the adversarial loss $L_{adv}$ encourages $F$ to learn domain-invariant features by confusing $D$, which can be achieved as follows:

$$L_{adv}(X_T) = -\sum_{j=1}^{N=H \times W} \sum_{k=1}^{K} sl^{(j,k)} \log(D(F(X_T)^{(j,k,0)}) \tag{7}$$

### 3.2.4. Multi-Level Adaptation Adversarial Framework

Although high-level features contain rich semantic information, there are equally critical contexts and spatial details in the low-level features, such as the position relation, contour information, and small-scale objects. Notably, it is significant for coastal HRRS images. In the background of segmentation models, integrating multi-level features has demonstrated an astonishing performance [42,43]. Motivated by these distinctive

approaches, we embed additional domain adaptation modules in the low-level feature stacks to enhance adaptability at multiple spatial scales. The overall objective function can be extended from Equations (6) and (7):

$$L(X_S, X_T) = \lambda_{seg} L_{seg}^S(X_S) + \sum_{i=1}^{n} \lambda_{adv}^i L_{adv}^i(X_T) \tag{8}$$

where $i$ denotes the level of feature stacks, $\lambda_{seg}$ and $\lambda_{adv}^i$ are the weights to balance the losses.

Our ultimate goal is to minimize the dense segmentation losses in $G = (F \rightarrow C)$ for both domains and maximize the probability of target features considered the source one. The min-max flow follows the formulation:

$$\min_{F,C} \max_{D} L(X_S, X_T) \tag{9}$$

*3.3. Implementation*

3.3.1. Subdivided Modules

Our CsCANet is built on the fully convolution architecture subdivided into a feature extractor, a pixel-level classifier, and four category-wise domain discriminators. It should be noted that the discriminators denote a similar structure with different channel numbers. Below, we elaborate their detailed compositional structures.

Feature extractor: According to our multi-level adaptation framework, the Resnet-101 [44] module pre-trained on the ImageNet [45] is adopted as the backbone network that works to extract features at multiple scales. The same as several advanced reports [3,30], we substitute for the down-sampling layers in the last two residual blocks with dilated convolutional layers. This strategy led to the size of the output feature map 1/8 of the input image, aiming to retain more spatial details without changing the scale of pre-trained parameters.

Pixel-level classifier: Referring to the Deeplab system [10,11], we leverage the ASPP module as an efficient pixel-level classifier that leverages four convolutional layers with a kernel size of $3 \times 3$ and a dilation of {6, 12, 18, 24} to form the network. The innovative structure successfully expands the receptive field to capture long-range context. For the module, the weights and biases are initialized with the Xavier [46] method.

Domain discriminators: We implement the category-space constraint domain adaptation with a category-wise discriminator. The network consists of three convolutional layers with a kernel size of $3 \times 3$ and channel numbers of $\{128 \times 2^i, 32 \times 2^i, N_C\}$, where $i = 1, 2, \dots, n$, presents the level of the adversarial learning scheme and $N_C$ gives the category number. Except for the last one, each convolutional layer is followed by the Leaky-ReLU [47] with a negative slope of 0.2. Besides, a bilinear up-sampling is used to reconstruct the resolution at the end of the discriminators. Additionally, similar to the classifier, we also use Xavier [46] to initialize the discriminators.

3.3.2. Training Details

In this section, our expected goal is to gain a well-trained adaptive segmentation network for CLCM. Alternate adversarial training is driven by the objective function $L$, in which the segmentation loss $L_{seg}^S$ and adversarial loss $L_{adv}^i$, respectively, serve for the dense prediction and multi-level domain adaptation task. The proper scheduling of these two modules is crucial for network performance. Thus, there is a greater weight for $L_{seg}^S$, i.e., $\lambda_{seg} = 1$. For the multi-level $L_{adv}^i$, we employ smaller weights for them, i.e., $\lambda_{adv}^i = \{0.0001, 0.0002, 0.0005, 0.001\}$, since the low-level features carry less semantic information.

Furthermore, to train our proposed CsCANet, we find that performing segmented training with self-supervised learning is an effective strategy to accelerate network parameter convergence. Within the front four-fifths of iterations, we begin by jointly training the source-based segmentation network and domain discriminators to conduce adversarial learning in one stage. In detail, source images are first forward to optimize $F$ and $C$ with

$L_{seg}^S$. The dense predictions $P_T$ are then generated from target images, which are forward put to $G$ and $D_i$ with the source labels $Y_S$ for optimizing $L_{adv}^i$. As for the rest of the iterations, the self-supervised learning adopts the generated pseudo target labels $Y_t$ to fine-turn the segmentation network for the target domain with $L_{seg}^T$. Algorithm 1 gives the necessary training process for our hybrid framework.

---

**Algorithm 1.** Training process for the hybrid framework.

---

**Input:** Source images $X_S$, source annotations $Y_S$, target images $X_T$, threshold $T$.
      Initialized feature extractor $F$, pixel-level classifier $C$, and discriminators $D_i$.
**Output:** Well-trained $F$, $C$, and $D_i$ for adversarial learning.
      Well-trained $F$ and $C$ for self-supervised learning.
**for** $k$ = 1 to max iterations ($M$) do
    **if** $k \leq 4/5\,M$ (adversarial learning)
        forward $X_S$, $Y_S$ to $F$ and $C$
        update $F$, $C$ with $L_{seg}^S$
        forward $X_T$ to $F$, $C$
        update predictions $P_T$ (score > $T$) = $T$
        forward $X_T$, $Y_S$, $P_T$ to $F$, $C$, $D_i$
        update $F$, $C$, $D_i$ with $L_{adv}^i$
        get pseudo label $Y_t$
    **else** $k > 4/5\,M$ (self-supervised learning)
        forward $X_T$, $Y_t$ to $F$ and $C$
        update $F$, $C$ with $L_{seg}^T$
    **end if**
**end for**

---

## 4. Experimental Results

### 4.1. Datasets Description

Two benchmark datasets, namely Shanghai and Zhejiang, are selected as the experimental data. As illustrated in Figure 4, corresponding study areas are located in typical coastal regions, and both of them are characterized by multi-scale land cover categories with low intra-class variances. Their appearances reflect the unique geographical characteristics of the coastal zone. Obviously, data diversity exists in spatial distribution between the Shanghai and Zhejiang datasets, where the former possesses more detailed information. Furthermore, due to the influences of seasonal factors and sensor modes, there are significant domain differences in spectral characteristics, which meets our experimental needs.

Shanghai dataset: The benchmark dataset is located in Xiaoshan District, Zhejiang Province, where the adopted remotely sensed images were acquired in 2017 with a spatial resolution of 0.8 m/pixel. In addition, the original images cover a scale of approximately 46 square kilometers with a spatial extent of 11,776 × 6144 pixels, composed of three bands of red (R), green (G), and blue (B). The images are further clipped into small patches with a size of 256 × 256 by employing a sliding window. As a result, there are 1104 images in the Shanghai dataset, where the ratio of training set to validation set is approximately 2:1 to the number of 736 and 368.

Zhejiang dataset: This dataset is located in Fengxian District, Shanghai, while the corresponding satellite images from the WorldView system were collected on 26 December 2016, with a high resolution of 0.5 m/pixel. It has been widely accepted that special ground objects in remote sensing images have a constant scale range [48]. Thus, the images are resampled to obtain a consistent spatial resolution as the Shanghai dataset. Besides, the same as the Shanghai dataset, the images only contain RGB channels and cover a region of approximately 61 square kilometers with a size of 12,800 × 7424 pixels. Similarly, the dataset contains 1450 images with a spatial extent of 256 × 256, and the numbers of training and validation sets are 967 and 483, respectively.

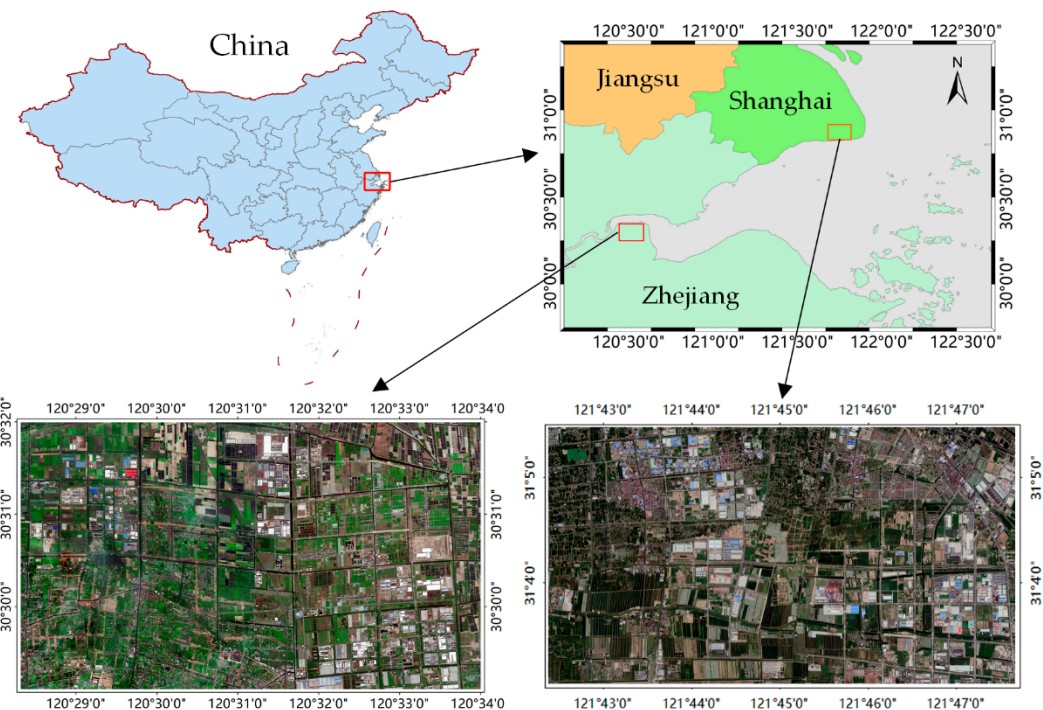

**Figure 4.** Overall presentation for our coastal datasets with respect to source locations and high-resolution remotely sensed images.

For both benchmark datasets, six land cover categories are defined and annotated at pixel-level (Figure 5). Specifically, the categories are composed of cropland (Cropland), impervious surfaces (Imp. Surf.), water areas (Water), vegetative cover (Veg.), bare land (Bareland), and roads (Road). Table 1 gives the pixel statistics of each dataset. It is hugely unbalanced for all the land cover categories that produce a more significant challenge to conduct dense segmentation work. As an example, the proportions of Cropland and Imp. Surf. are markedly larger than Bareland and Road.

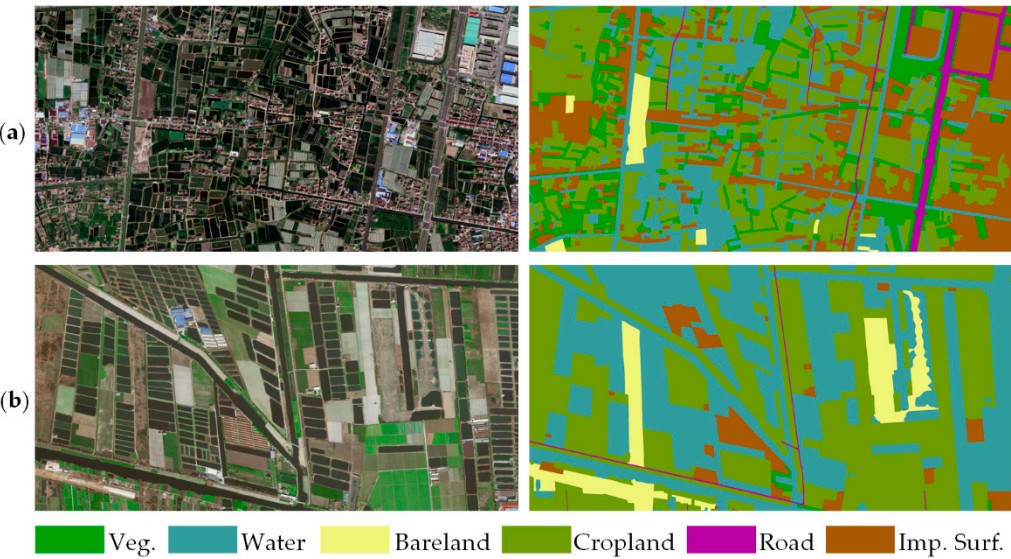

**Figure 5.** Representative examples of the Shanghai and Zhejiang datasets. (**a**) Images and corresponding annotated ground truth of the Shanghai dataset, (**b**) Images and corresponding annotated ground truth of the Zhejiang dataset.

**Table 1.** Pixel statistics among categories for Shanghai and Zhejiang datasets.

| Category | Shanghai Dataset | Zhejiang Dataset |
|---|---|---|
| Cropland | 32.29% | 46.03% |
| Imp. Surf. | 26.45% | 20.65% |
| Water | 11.44% | 13.46% |
| Veg. | 20.07% | 7.86% |
| Bareland | 3.96% | 7.77% |
| Road | 5.79% | 4.23% |

### 4.2. Experimental Setting

To verify the feasibility and robustness of the proposed method, two independent but opposite experiments are implemented on the above datasets. The unpaired images from the source and target domain are randomly taken as the network inputs, and the annotation is verifiable for the source data. In addition, data augmentation methods [49], i.e., mean subtraction and normalization, are leveraged on the training sets, which adjust the input images to accelerate the convergence of weights and biases.

We implement the proposed CsCANet on the PyTorch Toolbox [50] written as a deep learning framework. All experiments are conducted on a machine with an Intel Core i7-9700k (six cores), 16 GB of memory (RAM), an NVIDIA GeForce GTX 1080 GPU (8 GB), and an NVIDIA GeForce RTX 2080 GPU (8 GB). In the training procedure, we set 100 K iterations to obtain an overall convergence with a batch size of 1. For training the segmentation network, SGD [51] is used as the optimizer, whose momentum and weight decay are set by 0.9 and 0.0005. The learning rate is initialized to $2.5 \times 10^{-4}$ with a "poly" decay policy multiplied by $(1\text{-iter\_step/total\_step})^{0.9}$. To train the discriminators, we leverage the Adam [52] with momentums of 0.9 and 0.99 as the optimizer. The initial learning rate is $10^{-4}$ and decreases with the same policy as the segmentation network.

### 4.3. Evaluation Metrics

We take the CLCM with domain adaptation as a pixel-level and multi-category segmentation task, whose experimental results are generally evaluated via the generated confusion matrix. Referring to it, TP, FP, TN, and FN denote the numbers of true positives, false positives, true negatives, and false negatives [53,54]. Having access to these indexes, the following four metrics, i.e., per-class accuracy, overall accuracy, balanced F (F1) score, and intersection-over-union (IoU), are given to prove the validity effectiveness of our proposed CsCANet. Their detailed formulations are shown in Table 2, where $C$ presents the number of categories in both datasets. For all the metrics, the higher value demonstrates a better segmentation result to a certain degree.

**Table 2.** Evaluation metrics for the experimental results.

| Per-class Accuracy | $PA_c = \frac{TP_c}{TP_c + FP_c}$ |
|---|---|
| Overall Accuracy | $OA = \frac{1}{C} \sum_{c=1}^{C} \frac{TP_c + TN_c}{TP_c + TN_c + FP_c + FN_c}$ |
| Mean F1 Score | $precision = \frac{TP_c}{TP_c + FP_c}, recall = \frac{TP_c}{TP_c + FN_c}$ <br> $mF_1 = \frac{1}{C} \sum_{c=1}^{C} 2 \times \frac{precision \times recall}{precision + recall}$ |
| Mean IoU | $mIou = \frac{1}{C} \sum_{c=1}^{C} \frac{TP_c}{TP_c + FP_c + FN_c}$ |

### 4.4. Results and Analysis

In the experiments, several state-of-the-art methods devoted to cross-domain segmentation work are introduced and compared with our proposed CsCANet. They are driven by two different baseline networks: VGG16 [55] and ResNet101 [42], both of which are

pre-trained on ImageNet [43]. For the former, the representative models, i.e., FCNs in the wild (FCNs ITW) [20] and CyCADA [25], are presented. Instead, ResNet101-based others are provided by AdaptSegNet [21], CLAN [30], ADVENT [33], BDL [12], and FADA [35]. We perform fair comparisons on two adaptive CLCMs: Shanghai → Zhejiang and Zhejiang → Shanghai.

Partial representative examples from the above converse tasks are illustrated in Figures 6 and 7. Given as a pioneering work, FCNs ITW undoubtedly gains the worst results using a primitive feature alignment in the final representation layer (Figures 6a and 7a). It is apparent from Figures 6c and 7c that AdaptSegNet enables better performance than FCNs ITW by adopting the model in the output space. With pixel-level adaptation, the results of CyCADA and BDL are seriously affected by the early style transformation that results in the emergence of a large misclassification area, as shown in Figure 6b,f and Figure 7b,f. Even though CLAN and ADVENT further optimize the segmentation outcomes via the category-level adaptation and entropy minimization, there are still shortcomings in recognizing the ground objects with low inter-class variances (Figure 6d,e and Figure 7d,e). Besides, as shown in Figures 6h and 7h, FADA effectively solves the recognition issue of objects with similar characteristics. However, it presents poor ability in classifying the ground objects on a small scale, like other methods mentioned above. As we expected, the proposed method produces impressive segmentation results, as shown in Figures 6g and 7g. In practical terms, our CsCANet not only successfully recognizes the ground objects with similar appearance but also performs well in adapting the multi-scale features.

On the other hand, Tables 3 and 4 give the competitive evaluation metrics to other competitive methods on both segmentation tasks under domain adaptation, including per-class accuracy (PA), overall accuracy (OA), mean F1 score (mF1) and mean Iou (mIoU). The comparison results demonstrate that our proposed CsCANet achieves an excellent performance against other algorithms. CsCANet acquires the highest OA, mF1, and mIoU of 80.48%, 71.56%, and 57.69% in Shanghai → Zhejiang, while the corresponding values of 70.55%, 65.33%, and 49.64% also present the best ones in Zhejiang → Shanghai. Compared with the well-known category-level CLAN, CsCANet achieves 7.35% and 4.55% improvement in mIoU, which verifies the advance of our category-level adversarial framework. Furthermore, CsCANet still presents excellent representations on the per-class accuracy. For instance, it provides increases of 12.16% and 17.80% over the fine-grained FADA in Bareland and Road within the task Shanghai → Zhejiang.

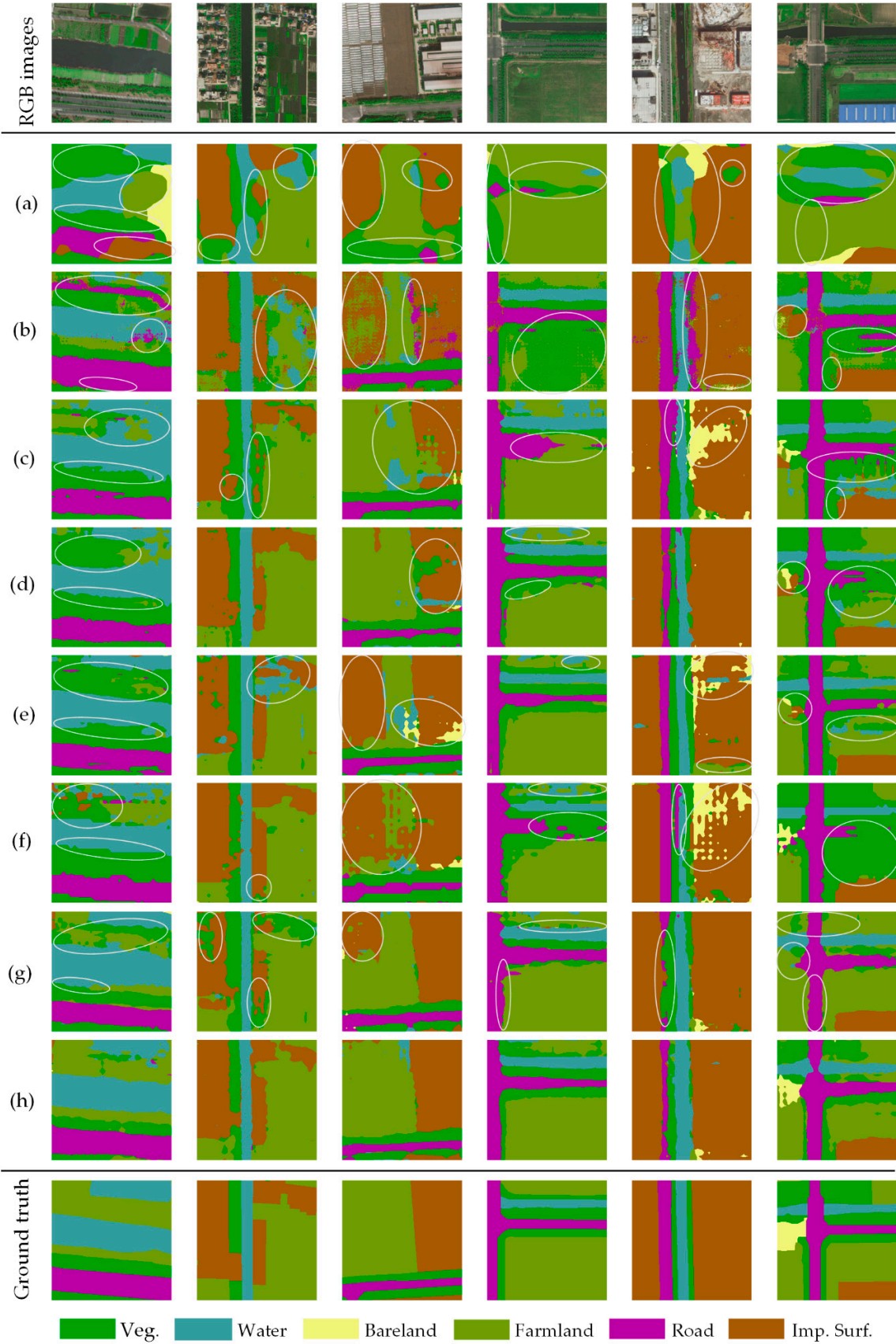

**Figure 6.** Representative examples of land cover mapping on the task Shanghai → Zhejiang: (**a**) FCNs ITW, (**b**) CyCADA, (**c**) AdaptSegNet, (**d**) CLAN, (**e**) ADVENT, (**f**) BDL, (**g**) FADA, (**h**) our CsCANet. RGB images and the corresponding ground truth are presented in unison. We circle the negative results in white ellipses.

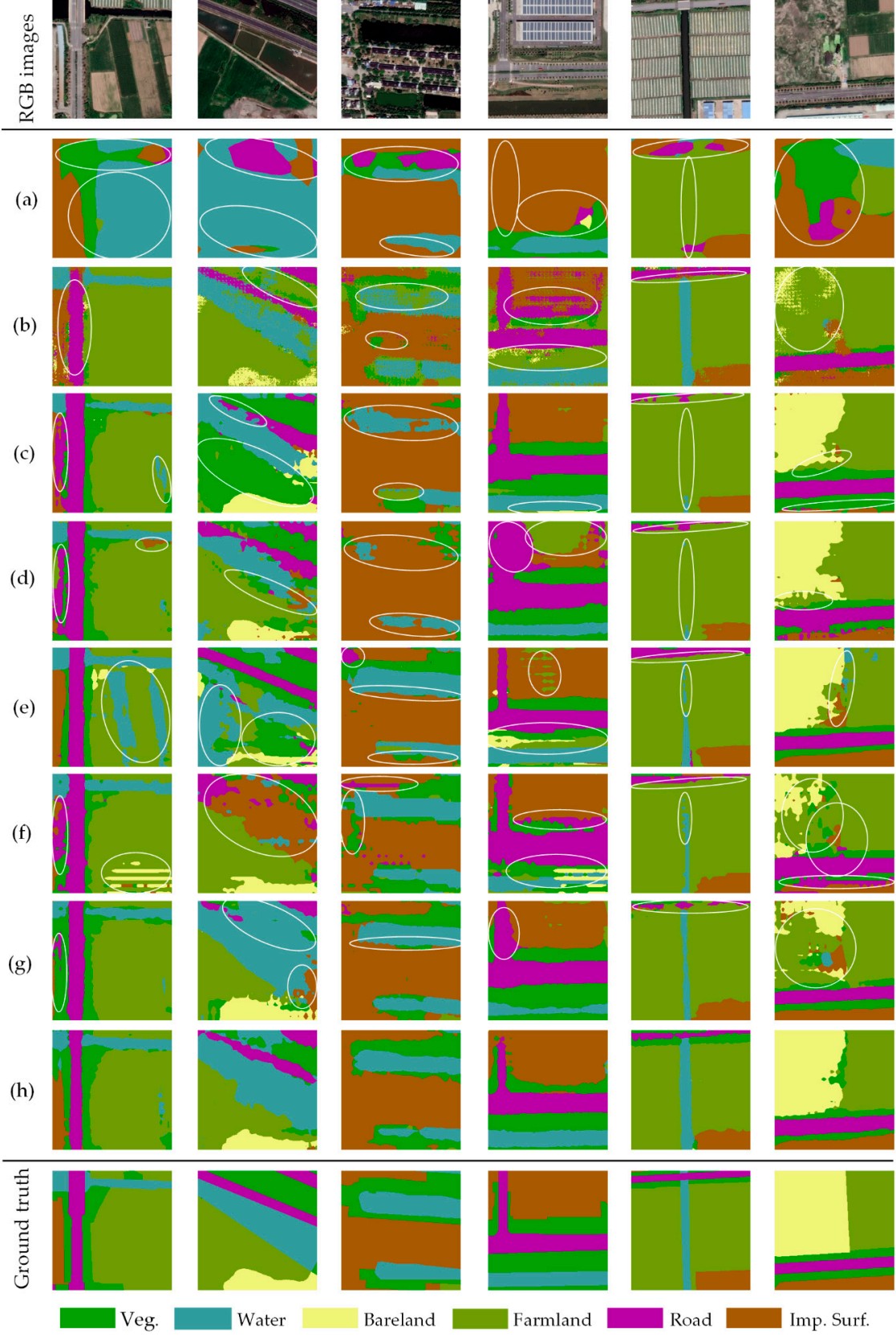

**Figure 7.** Representative examples of land cover mapping on the task Zhejiang → Shanghai: (**a**) FCNs ITW, (**b**) CyCADA, (**c**) AdaptSegNet, (**d**) CLAN, (**e**) ADVENT, (**f**) BDL, (**g**) FADA, (**h**) our CsCANet. RGB images and the corresponding ground truth are presented in unison. We circle the negative results in white ellipses.

**Table 3.** Comparison results on the task Shanghai → Zhejiang (%). We study the performance using the Shanghai dataset as annotated source data and the Zhejiang training set as the unlabeled target domain. The best results are highlighted in bold.

| | Shanghai → Zhejiang | | | | | | | | |
|---|---|---|---|---|---|---|---|---|---|
| Method | Veg. | Bare Land | Water | Imp.Surf. | Crop Land | Road | OA | mF1 | mIoU |
| FCNs ITW [20] | 40.93 | 12.43 | 40.56 | 59.18 | 79.20 | 10.16 | 58.76 | 40.96 | 28.52 |
| CyCADA [25] | 66.48 | 8.96 | 69.02 | 76.34 | 74.78 | 80.00 | 66.33 | 57.34 | 43.27 |
| AdaptSegNet [21] | 68.14 | 30.07 | 82.97 | 64.19 | 76.17 | 64.83 | 70.21 | 63.57 | 47.99 |
| CLAN [30] | **74.25** | 15.05 | 81.11 | 75.10 | 83.97 | 60.38 | 74.91 | 64.19 | 50.34 |
| ADVENT [33] | 69.77 | 33.05 | **89.13** | 69.43 | 81.55 | 66.45 | 75.03 | 67.90 | 52.94 |
| BDL [12] | 72.17 | 33.56 | 72.04 | 77.98 | 81.98 | 77.10 | 75.19 | 68.00 | 53.29 |
| FADA [35] | 59.88 | 27.32 | 80.98 | 74.62 | 89.89 | 63.98 | 77.42 | 68.15 | 53.93 |
| Our CsCANet | 42.41 | **39.48** | 76.50 | **80.92** | **94.69** | **81.46** | **80.48** | **71.56** | **57.69** |

**Table 4.** Comparison results on the task Zhejiang → Shanghai (%). We study the performance using the Zhejiang dataset as annotated source data and the Shanghai training set as the unlabeled target domain. The best results are highlighted in bold.

| | Zhejiang → Shanghai | | | | | | | | |
|---|---|---|---|---|---|---|---|---|---|
| Method | Veg. | Bare Land | Water | Imp. Surf. | Crop Land | Road | OA | mF1 | mIoU |
| FCNs ITW [20] | 21.26 | 17.89 | 53.24 | 90.26 | 25.47 | 5.43 | 43.18 | 31.91 | 20.57 |
| CyCADA [25] | 8.09 | 47.92 | 51.19 | 79.58 | 73.74 | 71.76 | 64.31 | 55.43 | 41.69 |
| AdaptSegNet [21] | 42.40 | **67.33** | 64.75 | 90.70 | 55.43 | 65.70 | 63.31 | 60.19 | 43.99 |
| CLAN [30] | 34.58 | 46.25 | 57.93 | 89.66 | 72.83 | 64.67 | 66.37 | 61.22 | 45.09 |
| ADVENT [33] | 46.57 | 44.29 | 59.48 | 89.36 | 57.30 | **77.36** | 64.37 | 61.23 | 45.48 |
| BDL [12] | 39.26 | 50.96 | 46.11 | 83.29 | 76.09 | 74.85 | 66.10 | 60.20 | 44.18 |
| FADA [35] | 36.98 | 30.25 | **71.88** | **94.81** | 67.55 | 66.03 | 67.44 | 61.61 | 46.13 |
| Our CsCANet | **48.35** | 38.75 | 60.62 | 86.53 | **78.26** | 74.63 | **70.55** | **65.33** | **49.64** |

## 5. Discussion

Whereas deep neural networks have driven the progress of land cover mapping, their performance fundamentally relies on the network architecture and optimization strategies. In this section, we conduct several ablation studies and effectiveness analysis for the above two primary factors. Note that all the comparative experiments are carried out on the CLCM task of Shanghai → Zhejiang.

### 5.1. Ablation Studies for Network Architecture

5.1.1. Design of Domain Adaptation Framework

Our anticipated goal is to develop a hybrid framework for land cover mapping across image domains. Considering the complex characteristics of coastal ground objects, we proposed a multi-level adaptation framework to adapt semantic features at different scales. Several comparative studies are executed to verify the effectiveness of our multi-level adversarial scheme. For all the methods, we use the pre-trained Deeplab v2 [9] as the baseline network.

As shown in Table 5, the domain adaptation approach achieves significant improvement in cross-domain land cover mapping. Compared with the baseline without any adaptation operation, our single-level CsCANet increases the value by 11.76% for mIoU. Additionally, the framework with multi-level generalization denotes significant results up to 77.52%, 69.25%, and 54.78% in terms of OA, mF1, and mIoU. On the other hand, Figure 8 visualizes partial representative outcomes. There is no doubt that the baseline method gives poor presentations because of the domain divergence within different datasets. In addition, multi-level CsCANet illustrates the better recognition ability in ground objects with low

intra-class variance and a small scale against the single-level approach. The aforementioned contrastive studies strongly prove the feasibility and effectiveness of the domain adaptation method in coastal land cover mapping, especially our multi-level adaptation framework.

**Table 5.** Performance comparisons of CsCANet with different domain adaptation frameworks in evaluation metrics. The best results are highlighted in bold.

| Method | Domain Adaptation | OA (%) | mF1 (%) | mIoU (%) |
|---|---|---|---|---|
| Baseline | No | 68.61 | 55.86 | 40.81 |
| CsCANet | Single-level | 76.76 | 67.44 | 52.57 |
| CsCANet | Multi-level | **77.52** | **69.25** | **54.78** |

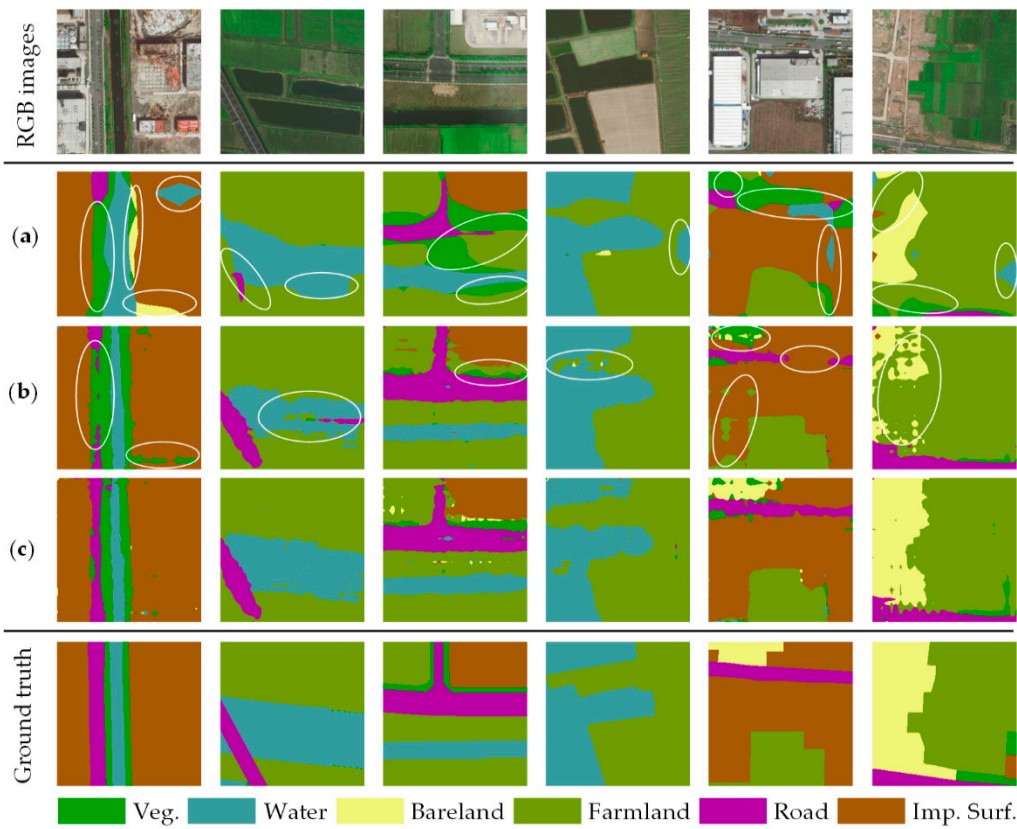

**Figure 8.** Representative examples of land cover mapping on the task Shanghai → Zhejiang: (**a**) baseline, (**b**) single-level CsCANet, (**c**) multi-level CsCANet. RGB images and the corresponding ground truth are presented in unison. We circle the negative results in white ellipses.

### 5.1.2. Design of Domain Labels Extraction Module

A conventional discriminator always employs a global adversarial loss to implement feature alignment via a binary domain label. Paying attention to the underlying category-space in the target domain, we introduced a category-wise discriminator with two summarized modules to extract domain labels. Moreover, a mixed strategy was leveraged in our method where the category-wise hard and soft labels were, respectively, applied to the source and target domain. In this subsection, we conduct ablation experiments to prove the superiority of our modules.

Table 6 gives the comparison results on different domain label extraction modules. The single strategy concerning category-wise hard and soft labels, respectively, achieves an increased mIoU of 2.83% and 2.65% compared to the naive one with a binary label. Notably, our data-based approach further improves network performance while achieving

the highest values of 77.52%, 69.25%, and 54.78% in the evaluation metrics. Meanwhile, Figure 9 offers specific references in per-class accuracy as additional verifications. As can be seen, the mixed method presents better results in most categories, such as the Bareland, Road, Cropland, etc. In summary, extracting domain labels with reasonable modules can lead to outstanding presentation for network performance, although it is determined by the data structure itself.

**Table 6.** Performance comparisons of CsCANet with different domain label extraction modules in evaluation metrics. *S* and *T* present the source and target domain. The best results are highlighted in bold.

| Method | Binary Label | Category-Wise Hard Label | Category-Wise Soft Label | OA (%) | mF1 (%) | mIoU (%) |
|---|---|---|---|---|---|---|
| CsCANet | *S*, *T* | | | 74.43 | 66.00 | 50.75 |
| CsCANet | | *S*, *T* | | 76.93 | 68.17 | 53.58 |
| CsCANet | | | *S*, *T* | 76.40 | 68.26 | 53.40 |
| CsCANet | | *S* | *T* | **77.52** | **69.25** | **54.78** |

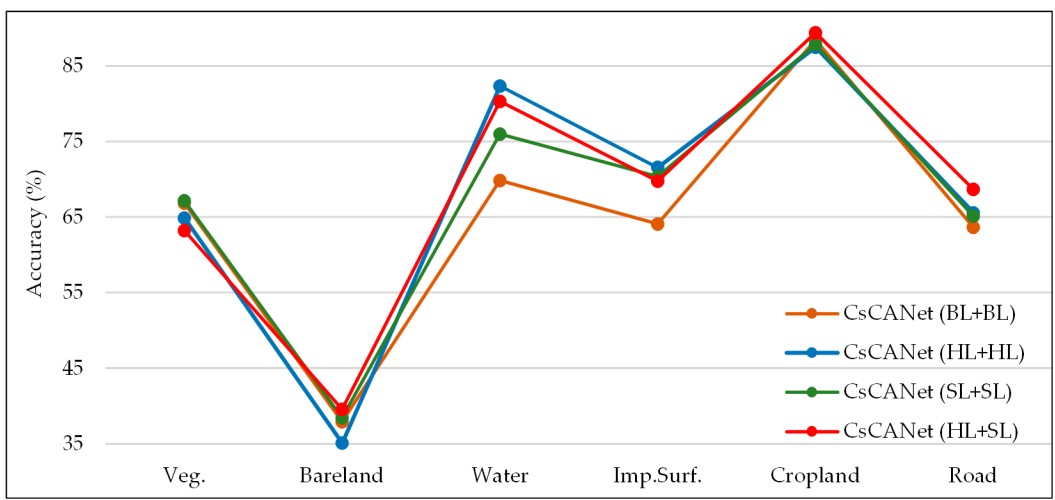

**Figure 9.** Per-class accuracies for land cover mapping. BL, HL, and SL, respectively, denote the binary label, category-wise hard label, and soft label. For all the categories, each method with a specific extraction module for domain labels corresponds to a broken line with a particular color.

### 5.2. Effectiveness Analysis of Improvement Strategy

To further optimize the segmentation results, we employed the self-supervised learning approach as an improvement strategy to perform segmented training. Therefore, two comparative experiments under contrary settings are carried out to analyze its effectiveness. They are based on our multi-level adaptation framework and mixed approach to domain label extraction.

The comparison results from Table 7 confirm that self-supervised learning led to a remarkable improvement, where the values of OA, mF1, and mIoU increased by 2.96%, 2.31%, and 2.91%. We also illustrate the internal specific variation of IoU for each category, as shown in Figure 10. Compared with the original one, it is apparent that the CsCANet with self-supervised learning presents further improvements to all the object categories, especially the Imp. Surf., Road and Bareland. These pieces of ample evidence indicate the effectiveness of our improvement strategy with self-supervised learning.

**Table 7.** Performance comparisons of CsCANet with different improvement strategies in evaluation metrics. The best results are highlighted in bold.

| Method | Self-Supervised Learning | OA (%) | mF1 (%) | mIoU (%) |
|---|---|---|---|---|
| CsCANet | | 77.52 | 69.25 | 54.78 |
| CsCANet | √ | **80.48** | **71.56** | **57.69** |

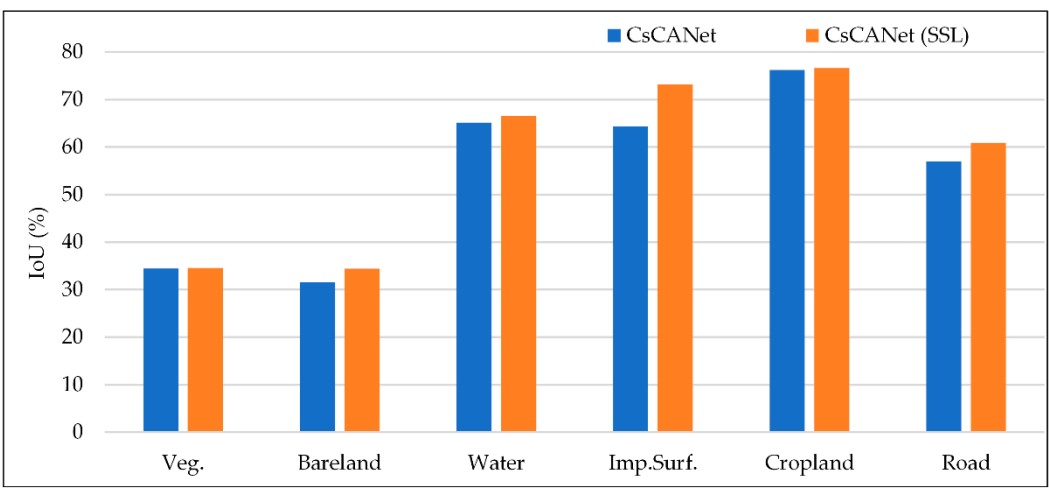

**Figure 10.** Per-class intersection-over-union (IoU) for land cover mapping. For all the categories, each method corresponds to a histogram with a particular color.

## 6. Conclusions

This paper proposes a novel category-level adaptative method to address the cross-domain CLCM with HRRS images. We take the adversarial framework with a category-wise discriminator as an alternative to the conventional one, then generalize it to multiple levels. Several state-of-the-art models are employed and compared to verify the superiority of the proposed method. The experimental results demonstrate that our approach successfully learns the transformed features and executes the domain adaptation procedure. In addition, the multi-level adversarial scheme is confirmed to be efficient in recognizing the ground objects with low intra-class variances and spatial details, ultimately achieving the optimal performance in adaptive pixel-level segmentation. Furthermore, massive ablation studies strongly confirm the effectiveness of our network architecture and improvement strategy. Nevertheless, our method merely takes the dense annotations from the source domain as supervised guidance. In future research, we will focus on other effective guidance information, such as the semantic context and super-pixel, to further improve the network performance by implementing additional constraints.

**Author Contributions:** Conceptualization, J.C. and G.Z.; methodology, J.C. and G.C.; software, J.C.; validation, N.Y. and P.Z.; formal analysis, B.F.; investigation, P.Z.; resources, B.F.; data curation, P.Z.; writing—original draft preparation, J.C.; writing—review and editing, G.C. and B.F.; visualization, N.Y.; supervision, G.Z.; project administration, J.C.; funding acquisition, G.C. All authors have read and agreed to the published version of the manuscript.

**Funding:** The project was funded by the National Natural Science Foundation of China under Grant No. 41674015 and the Scientific Research Project of Hubei Province under Grant 1232039.

**Institutional Review Board Statement:** Not applicable.

**Informed Consent Statement:** Not applicable.

**Data Availability Statement:** The data presented in this work are available on request from the corresponding author. The data are not publicly available due to other ongoing studies.

**Acknowledgments:** The authors would like to acknowledge the following organization, the Hubei Key Laboratory of Marine Geological Resources (MGR202005), for partly funding this study. The authors also thank the anonymous reviewers for their constructive suggestions.

**Conflicts of Interest:** The authors declare no conflict of interest.

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
