# Peer review of "Unsupervised Domain Adaption for High-Resolution Coastal Land Cover Mapping with Category-Space Constrained Adversarial Network"

_remotesensing, doi:10.3390/rs13081493_

Round 1

Reviewer 1 Report

This paper presents interesting results of a methodology to achieve land cover mapping using a  novel category-space constrained adversarial network, which is a tool that is increasingly being used in numerous research projects. The article explains not only the methodology but also includes a comparison with other state-of-the-art models. However, the authors should kindly consider some minor changes before publication for a potential improvement. The article is quite interesting, so I recommend the publication.

Line 361 and 369

One dataset has a spatial resolution of 0.8 m/pixel, and the other 0.5 m/pixel.

Did the authors harmonize the data? If not, could this fact affect the results? Why? Explain it in the discussion, please.

Line 216 and 413

It is not clear what is the validation method. Even a knowledgeable reader may encounter difficulties since there are many different methodologies. Please explain it clearly.

Line 440

How did the authors have to obtain the ground truth data? Visually? Did the authors survey the zone? Please explain it.

Line 525

What is, theoretically, the explanation for this remarkable improvement? Effectively, the results are precise, but why?

Reviewer 2 Report

The manuscript “Unsupervised Domain Adaption for High-Resolution Coastal Land Cover Mapping with Category-Space Constrained Adversarial Network” submitted by Jifa Chen, Guojun Zhai, Gang Chen, Bo Fang, Ping Zhou, and Nan Yu is an interesting and well written paper.

By considering the proposed methodology, its descriptions and content and the descriptions of the results, I would recommend acceptance for publication of this manuscript after some minor changes.

Below I present some points to be taken into account in order to enhance the overall quality of the manuscript.

Lines 34-36: What do you mean by “land-sea junction “? Please rephrase  because this might be interpreted as a shoreline.    

Line 41: What do you mean by “acquired in the wild “? Please rephrase.

Line 130: Please correct “since Goodfellow [19] proposed“ to “since Goodfellow et al. [19] proposed“. Please check for the whole manuscript for other citations.

Line 360: How the data were acquired? Please provide details on the sensor, platform and acquisition method and time of acquisition.

Line 368: Please provide the name of Satellite and the time of acquisition

Lines 439, 443: Please provide the scale for the images used for the Figures 6 and 7.

Lines 484, 485, 486:  How large the area of study could be to conduct the “coastal land cover mapping“ using the proposed methodology?    
